# Generation of Wheat Near-Isogenic Lines Overexpressing 1Bx7 Glutenin with Increased Protein Contents and SDS-Sedimentation Values

**DOI:** 10.3390/plants12061244

**Published:** 2023-03-09

**Authors:** Myoung-Hui Lee, Changhyun Choi, Kyeong-Hoon Kim, Jae-Han Son, Go-Eun Lee, Jun-Yong Choi, Chon-Sik Kang, Jiyoung Shon, Jong-Min Ko, Kyeong-Min Kim

**Affiliations:** Wheat Research Team, National Institute of Crop Science, Rural Development Administration, Wanju 55365, Republic of Korea

**Keywords:** hexaploid wheat, 1Bx7^OE^, 1Bx7 promoter, ultra-performance liquid chromatography, near infrared reflectance spectroscopy

## Abstract

Overexpression of Glu-1Bx7 via allele 1Bx7^OE^ significantly contributes to high dough strength in some wheat varieties and is useful for improving wheat quality. However, the proportion of wheat varieties containing Bx7^OE^ is quite low. In this study, four cultivars containing 1Bx7^OE^ were selected, and among the selected varieties, Chisholm (1Ax2*, 1Bx7^OE^ + 1By8*, and 1Dx5 + 1Dx10) was crossed with Keumkang, a wheat variety that contains 1Bx7 (1Ax2*, 1Bx7 + 1By8, and 1Dx5 + 1Dx10). SDS-PAGE and UPLC analyses showed that the expression of the high-molecular-weight glutenin subunit (HMW-GS) 1Bx7 was significantly higher in NILs (1Ax2*, 1Bx7^OE^ + 1By8*, and 1Dx5 + 1Dx10) compared with that in Keumkang. Wheat quality was analyzed with near infrared reflectance spectroscopy by measuring the protein content and SDS-sedimentation of NILs. The protein content of NILs (12.94%) was 21.65% higher than that of Chisholm (10.63%) and 4.54% higher than that of Keumkang (12.37%). In addition, the SDS-sedimentation value of NILs (44.29 mL) was 14.97% and 16.44% higher than that of Keumkang (38.52 mL) and Chisholm (38.03 mL), respectively. This study predicts that the quality of domestic wheat can be improved by crossbreeding with 1Bx7^OE^-containing cultivars.

## 1. Introduction

Wheat (*Triticum aestivum* L.) is a hexaploid (2*n* = 6x = 42, AABBDD) and a major crop grown worldwide as a flour source for various foods. According to their solubility, wheat grain proteins can be classified into four types: albumin, globulin, gliadin, and glutenin [1]. Proteins in wheat grains, especially gluten proteins, are generally considered to be the most important factor in determining the dough properties and bread quality [2,3,4]. Gluten is the largest natural polymer composed of disulfide bonds and non-covalent hydrogen bonds between glutenin and gliadin [5]. Gliadins are responsible for the extensibility and viscosity of dough [6], whereas glutenin affects the strength and elasticity of wheat dough [7]. There are two major types of glutenin, high-molecular-weight glutenin subunits (HMW-GS) and low-molecular-weight glutenin subunits (LMW-GS). HMW-GS greatly influence the end-use quality of wheat [6,8] and constitute linear chains and protein networks. On the other hand, LMW-GSs exist as clusters and aggregates formed by branching from the linear chains. The HMW-GSs are encoded by *Glu-A1*, *Glu-B1*, and *Glu-D1* alleles located at the *Glu-1* loci on the long arms of group 1 chromosomes 1A, 1B, and 1D of hexaploid wheat, respectively [7]. Each locus consists of two tightly linked genes encoding the x- and y-type subunits [6]. In sodium dodecyl sulfate-polyacrylamide gel electrophoresis (SDS-PAGE), the x-type subunit has a relatively high molecular weight, and the y-type subunit has a relatively high electrophoretic mobility [9,10,11,12,13,14]. Theoretically, hexaploid common wheat should have six HMW-GS (*1Ax*, *1Ay*, *1Bx*, *1By*, *1Dx*, and *1Dy*). Almost all hexaploid cultivars, with the exception of a few that express all six HMW-GSs, express three to five HMW-GS [15,16,17], and the *1Ay* subunit is silent at a high frequency [6]. The *Glu-1* locus exhibits a high level of genetic polymorphism [18], and more than 100 allelic variations have been identified [19]. In particular, *Glu-1B* and *Glu-1D* have many variants, whereas *Glu-1A* is less variable.

Alleles at *Glu-1B* encode three homologous x-type subunits designated 1Bx7, 1Bx7*, and 1Bx7^OE^. Overexpression of Glu-1Bx7 by allele 1Bx7^OE^ contributes to high dough strength in some varieties [20,21]. The absence of 1Bx7 negatively affects sponge cake performance because it negatively affects the microstructure of the gluten network [22]. In contrast, 1Bx7^OE^ alters the secondary structure and microstructure of gluten to improve the mixing and rheological properties of dough [23] and confers superior baking quality compared to that of wheat cultivars with the 1Bx7 subunit [24,25,26,27]. Several large insertions or deletions (InDels), such as a 43 bp InDel at −1047, a 54 bp duplication at −400 upstream, and a 185 bp miniature inverted-repeat transposable element (MITE) at −874, resulted in major divergence among four promoters (1Bx7, 1Bx7^OE^, 1Bx13, and 1Bx14), and promoter sequence variation led to differential expression [28]. The 54 bp tandem duplication was absent in non-Glu-1Bx promoters but was present in all Glu-1Bx promoters except for Glu-1Bx13. The 43 bp insertion is strongly associated with high 1Bx7^OE^ expression [29]. The 43 bp insertion is present not only in the 1Bx7 promoter but also in other 1Bx promoters, such as 1Bx14 and 1Bx6. However, the 43 bp insertion in 1Bx6 and 1Bx14 did not lead to overexpression at the protein level [28]. Geng et al. (2014) suggested that co-regulatory factors linking the 43 bp insertion with expression efficiency are present in the 1Bx7 allele and that regulation at the translational level may strongly influence the differential expression between 1Bx7 and non-1Bx7 subunits [28]. The 185 bp MITE insertion was present in both the Glu-1Bx14 and Glu-1Bx20 promoters, which may only slightly affect transcriptional regulation. GUS expression directed by Pro-1Bx14 harboring the 185 bp MITE was higher than that directed by Pro-1Bx14 lacking the 185 bp MITE [28], but this insertion had little effect on gene expression [1,30,31]. It has been reported that the 1Bx7 subunit is overexpressed due to duplication of a 10.3-kb segment containing the *1Bx7* gene and a flanking long terminal repeat (LTR) retroelement, as well as transcriptional regulation in some cultivars [32,33]. Geng et al. reported that a cultivar with the 43 bp insertion showed higher expression of 1Bx7 than the control, whereas a cultivar with the 43 bp InDel and two copies of *1Bx7* (gene duplication) showed the highest protein level [28].

The contribution of the 1Bx7^OE^ subunit to the formation of stronger dough can provide useful information for improving wheat quality. Therefore, breeding more wheat varieties with the 1Bx7^OE^ subunit would be a promising way to improve wheat quality. However, the proportion of wheat varieties containing 1Bx7^OE^ is low. In addition, crossbreeding takes a long time. In this study, wheat varieties overexpressing 1Bx7 were selected using promoter and protein analyses. Next, the selected cultivar Chisholm was crossed with Keumkang, a domestic wheat cultivar, using a speed-breeding method. Finally, after the selection of homozygous Keumkang near isogenic lines (NILs), the expression of 1Bx7 was evaluated using SDS-PAGE and ultra-performance liquid chromatography (UPLC). In addition, the protein content and SDS-sedimentation of the NILs were measured to determine wheat quality by near infrared reflectance spectroscopy analysis. These results are expected to be useful in wheat breeding programs targeting high-quality, strong flour.

## 2. Results

### 2.1. Selection of Bx7^OE^ in Common Wheat Cultivars and Evaluation of 1Bx7 Expression Levels Using UPLC

We selected 43 bp insertions and duplications in *1Bx7* varieties from 607 wheat accessions of the National Institute of Crop Science in the Republic of Korea (Table 1, Figure 1) by using selection markers (Appendix A). Three varieties (Tamaizumi, MT8195, and K2244) had only 43 bp insertions in the promoter region, and four varieties (MT8191, Chisholm, KS85WGRC01, and GR863) had duplicated 10.3-kb segments containing the *1Bx7* gene and a flanking long terminal repeat (LTR) retroelement (Figure 1).

The composition and expression levels of different HMW-GSs of the seven varieties with the 43 bp InDel were compared to those of Keumkang, a Korean wheat variety, using UPLC (Figure 2) and SDS-PAGE (Appendix A). Keumkang, a triple hybrid of Geurumil/Kanto 75/Eunpamil, has a fast growth rate, a high milling rate (75%), and a hard white grain color [34]. This variety has the following compositions: 1Ax2*, 1Bx7 + 1By8, and 1Dx5 + 1Dx10 HMW-GS. However, Keumkang lacks the 43 bp insertion in the 1Bx7 promoter. The proportion of 1Bx7 subunits was quantified using the area ratio (%) of the 1Bx7 subunit in the elution profiles to the total amount of HMW-GS. The 1Bx7 ratio of the total HMW-GS ratio was 1.22-, 1.30-, and 1.39-fold higher in Tamaizumi, MT8195, and K2244, respectively, than in Keumkang. However, the 1Bx7 ratio of the total HMW-GS ratio was 1.71-, 1.74-, 1.68-, and 1.67-fold higher in MT8191, Chisholm, KS85WGRC01, and GR863, respectively, than in Keumkang (Figure 2). In addition, the peak heights of Tamaizumi (~183 mAU), MT8195 (~206 mAU), and K2244 (~179 mAU) were not significantly different from those of Keumkang (~220 mAU). Whereas, those of MT8191 (356 mAU), Chisholm (455 mAU), KS85WGRC01 (417 mAU), and GR863 (441 mAU) were significantly higher than those of Keumkang (supporting data can be found in Appendix A).

### 2.2. Evaluation of the Expression Level of 1Bx7 Using Various 1Bx Promoters

We investigated the level of *1Bx7* expression in Keumkang seeds at various stages using reverse transcription PCR. The *1Bx7* transcript was first detected at 21 days after flowering (DAF) (Appendix A). Furthermore, we generated GFP constructs harboring the 43 bp insertion and control promoter (Figure 3A).

These constructs were transfected using *Agrobacterium*-mediated transformation of immature Keumkang grains 35–40 days after heading (DAH). The images of the seed maturation stages after heading are shown in Appendix A. The GFP expression of the 43 bp insertion (hereinafter referred to as the “43+”) promoter was normalized to the expression of PAT contained in the vector [35,36] and compared with that of the control (hereinafter referred to as the “43−”) promoter. GFP expression was higher in the 43+ promoter than in the 43− promoter (Figure 3B). These results support previously reported findings that 43+ affects the regulation of 1Bx7 expression [28,31].

### 2.3. Introduction of the 1Bx7^OE^ Subunit into Domestic Wheat Varieties through Speed Breeding

Among the selected 1Bx7^OE^ varieties, Chisholm was crossed with Keumkang and backcrossed three times with Keumkang. Two, ten, and five plants with 6, 32, and 27 seeds from BC_1_F_1_ to BC_3_F_1_ were screened using the MAR marker (Appendix A). The presence of 1Bx7^OE^ was confirmed in seven of the 37 BC_3_F_2_ plants, and 1Bx7^OE^ homozygous lines were selected from the BC_3_F_3_ plants. The breeding scheme of Keumkang NILs (hereinafter referred to as NILs) is shown in Figure 4. As for the length and width of the seeds, Chisholm was longer than Keumkang. The seed size and seed coat color of NILs were similar to those of Keumkang, the recurrent parent, but different from those of Chisholm (Appendix A).

### 2.4. Evaluation of 1Bx7^OE^ Expression in NILs

To identify the different HMW-GS compositions and their expression levels in the NILs, SDS-PAGE and UPLC analyses were conducted on BC_3_F_3_ homozygous lines. SDS-PAGE analysis showed that Keumkang and Chisholm contain 1Bx7 (43−) and 1Bx7^OE^, respectively. In the NILs, the 1Bx7 band was thicker than that of Keumkang and similar to that of Chisholm (Figure 5). To confirm the expression levels of individual HMW-GS, UPLC was performed to determine the ratio of individual HMW-GS to total HMW-GS (supporting data can be found in Appendix A). The 1Bx7^OE^ showed a peak height that was 1.6–1.7-fold higher than that of Keumkang and similar to that of Chisholm (Figure 6). In addition, both SDS-PAGE and UPLC results showed that NILs had a similar LMW-GS composition to that of Keumkang (Figure 2, Figure 5 and Figure 6).

### 2.5. Wheat Quality Analysis Using near Infrared Reflectance Spectroscopy (NIRS)

To determine wheat quality, protein content and SDS-sedimentation were determined by NIRS [37]. The average wheat protein contents of NILs, Chisholm, and Keumkang were 12.94%, 10.63%, and 12.37%, respectively. The average protein content of NILs was 21.65% higher than that of Chisholm and 4.54% higher than that of Keumkang (12.37%). The average wheat SDS-sedimentation values of NILs, Chisholm, and Keumkang were 44.29 mL, 38.03 mL, and 38.52 mL, respectively. The average SDS-sedimentation values were 16.44% and 14.97% higher than those of Chisholm and Keumkang, respectively (Figure 7).

## 3. Discussion

Two factors causing 1Bx7 overexpression have been reported, namely, a 43 bp insertion in the promoter and gene duplication, but the effects are controversial. Butow et al. (2004) and Ragupathy et al. (2008) suggested that the gene duplication was associated with overexpression of 1Bx7 [29,33] and that the 43 bp insertion was not critical for overexpression of the 1Bx7 subunit [29]. 1Bx7^OE^ lines contain the 43 bp insertion, but the reverse is not always true [33]. However, Geng et al. (2014) reported that 1Bx7 is regulated by both gene duplication and transcription [28]. However, the 43 bp insertion exists not only in the 1Bx7 promoter but also in other 1Bx promoters, such as those of 1Bx14 and 1Bx6, but is not significantly associated with an increase in protein levels [28]. The factors regulating 1Bx7, 1Bx6, and 1Bx14 have not yet been characterized. In this study, we used seven varieties containing the 43 bp insertion in the promoter region with or without gene duplication. Similar to the results of Ragupathy et al. (2008), SDS-PAGE and UPLC results showed that the 1Bx7^OE^ phenotype included both the 43 bp insertions in the promoter and gene duplication. The ratio of 1Bx7 to total HMW-GS slightly increased in the cultivars with only the 43 bp insertion in the promoter without gene duplication compared to Keumkang, in which the 43 bp was not inserted, but the ratio was much lower than that of the gene duplication. These results support the thesis that both the 43 bp insertion in the promoter and *1Bx7* duplication are important for 1Bx7 overexpression.

The HMW-GS are highly related to end-use quality and account for approximately 10% of the total mass of gluten proteins [38]. Most varieties with gluten properties suitable for bread have 1Ax1 or 1Ax2*, 1Bx17 + 1By18 or 1Bx7^OE^ + 1By8, and 1Dx5 + 1Dy10 at the *Glu-A1*, *Glu-B1*, and *Glu-D1* loci, respectively. In particular, varieties with the 1Bx7^OE^+1By8 allele have excellent baking properties [39,40]. Wheat cultivars containing 1Bx7^OE^ are rare in Asia, including Korea, and many Korean wheat cultivars have the *Glu-D1f* (1Dx2.2 + 1Dy12) allele [41]. In contrast, 1Bx7^OE^ is widely distributed in Australia, the USA, and Europe [33]. Although the domestic cultivar Jaeraejon Mil (IT166460) containing 1Bx7^OE^ has been reported, the composition of HMW-GS of Jaeraejon Mil is 1Ax1 (*Glu-A1a*), 1Bx7^OE^ + 1By8 (*Glu-B1al*), and 1Dx2 + 1Dy12 (*Glu-D1a*). The 1Bx7 expression in Jaeraejon Mil is high, but it contains 1Dx2 + 1Dy12 instead of 1Dx5 + 1Dy10 (*Glu-D1d*) [42]. Park et al. (2006) reported that bread made with Korean wheat flour was not only less bulky than bread made using commercially available breadcrumbs but also that bread made with Korean wheat bearing the *Glu-D1d* allele was of poor quality [43]. Compared with the normal 1Bx7 subunit, the 1Bx7^OE^ subunit of a wheat NIL has been reported to increase the content of β-sheets in the gluten secondary structure and to be associated with superior rheological properties [23]. This suggests that cultivar improvement by transferring the desired *Glu-1B* allele to domestically cultivated wheat via crossbreeding is necessary. In the four selected strains, 1By protein not only showed a release time similar to that of 1Bx9 in UPLC analysis but was also very similar in protein size to 1Bx8 in SDS-PAGE (Appendix A). In a previous study, Butow et al. (2004) reported that 1By8 and 1By8* had similar electrophoretic mobilities and could not be clearly distinguished using SDS-PAGE. Reverse-phase high-performance liquid chromatography is required to distinguish 1By8* from 1By8 [44]. This suggests that the 1By protein of the four selected cultivars was 1Bx8*. In this study, the HMW-GSs NILSs had Glu-1Ax2*, 1Bx7^OE^ + 1By8*, and 1Dx5 + 1Dy10 was included by crossing Keumkang and Chisholm. The *Glu-B1b*+*2o*/*2a*-encoding subunits Bx7*+By8/By8* exhibited positive effects on bread-making quality. When the *Glu-1A* allele was either *Glu-Ala* (Ax1) or *Glu-A1b* (Ax2*), the Glu-score of *Glu-B1b*+*2o* and *Glu-B1b*+*2a* was 3, which had a similar effect on quality. These results suggest that By8 and By8* have similar effects on quality [45]. The BC_3_F_3_ generation, LMW-GS, showed the composition of Keumkang rather than that of Chisholm. Some lines with compositions 1Ax2*, 1Bx7^OE^ + 1By8, and 1Dx5 + 1Dy10 were detected, but they were heterozygous, and 1Bx7^OE^ + 1By8 was not detected as homozygous. The reason the 1Bx7^OE^ + 1By8 combination could not be found was that the x- and y-type alleles are closely linked and do not recombine. Another possibility is that there were not sufficient selected lines.

The protein content of Keumkang was higher than that of Chisholm and slightly higher than that of Keumkang in NILs. The protein content of the NILs seemed to increase slightly after introducing Bx7^OE^ in Keumkang, but not significantly. Although the protein content of Chisholm was lower than that of Keumkang, the SDS-sedimentation value of Chisholm was similar to that of Keumgang because of the overexpression of Bx7. In addition, the overexpression of Bx7 in NILs made the SDS-sedimentation value higher than that of both Keumgang and Chisholm. As a result, NILs containing 1Bx7^OE^ are expected to improve wheat quality compared to Keumkang, but further studies are needed to determine whether the baking properties of NILs are improved over Keumkang. These results are expected to be useful for wheat breeding programs targeting high-quality, strong flour.

## 4. Materials and Methods

### 4.1. Plant Materials

A total of 607 accessions of hexaploid wheat (*Triticum aestivum* L.) from the Korean National Institute of Crop Science (Wanju, Republic of Korea) were germinated on Petri dishes with moist paper. The wheat varieties were provided by the Korean Agricultural Culture Collection (KACC) (http://genebank.rda.go.kr, accessed on 9 January 2023). Genomic DNA was extracted from leaf tissue using a Higene™ Genomic DNA Prep kit (solution type; BioFACT, Daejeon, Republic of Korea). 1Bx7 and 1Bx7^OE^ were identified using specific DNA markers (MAR) [29] and left and right junction primers [33]. PCR was performed under the following conditions: 15 min at 95 °C, 35 cycles of 30 s at 94 °C, 30 s at 55 °C, and 30 s at 72 °C on a thermal cycler (Applied Biosystem, MA, USA) with 20 μL reaction volumes for each sample containing 50 ng template DNA, 10 pmole of each primer, a final 1× reaction buffer, 250 μΜ dNTPs, and 1 unit of Taq polymerase (Genetbio Prime DNA polymerase, Republic of Korea). The PCR products were electrophoresed on an 1.2% agarose gel and visualized using a Davinch-K gel imaging system (Davinch-K, Seoul, Republic of Korea). Primers are listed in Appendix A.

### 4.2. Expression of 1Bx7^OE^ at Different Times after Flowering (DAF) and Reverse Transcription PCR

RNA was extracted from immature wheat seeds at 7, 14, and 21 DAFs by using the TRIzol (Thermo Scientific, USA) method, while the genomic DNA was removed using a TURBO DNA-Free Kit (Thermo Scientific). cDNA was synthesized using a high-capacity cDNA reverse transcription kit (Thermo Scientific). RT-PCR was performed under the following conditions: 15 min at 95 °C, 25 cycles of 30 s at 94 °C, 30 s at 55 °C, and 30 s at 72 °C on a thermal cycler (Applied Biosystems, MA, USA). The PCR products were loaded onto a 1.2% agarose gel. The primers used for PCR were as follows: 5′-ATGCCAACAGGTGGTGGACC-3′ and 5′-AAGTTACACTTGGGTAATAC-3′ for *1Bx7*; 5′-GCCACACTGTTCCAATCTAT-3′ and 5′-TGATGGAATTGTATGTCGCTTC-3′ for actin. Actin was used as the internal control.

### 4.3. Construction of proGlu-1Bx7:GFP for the Promoter Transient Expression Assay

Genomic DNA was extracted from leaf tissue using the Higene™ Genomic DNA Prep kit. Using a pair of specific primer sets, two Glu-1Bx promoters of a length of 1290 and 1247 bp for the 43 bp insertion and control promoter regions were amplified in the wheat varieties Chisholm and Chinese Spring, respectively. The primers used for PCR were as follows: 5′-CCTCAGCATGCAAACATGCA-3′ and 5′-CTCAGTGAACTGTCAGTGAA-3′. PCR amplification products were purified using a QIAquick Gel Extraction Kit (Qiagen, Germany) and ligated into pENTR/D-TOPO (Invitrogen, USA) to create entry vectors. The entry vectors, which were confirmed by sequencing analysis, were subjected to LR reactions (recombination between *att*L and *att*R sites) with the pBGWFS7 destination vector containing GFP to generate pBGWFS7-*1Bx7:GFP*.

### 4.4. Quantification of Promoter Expression by Western Blot Analysis

Immature wheat grains at 40 days after heading were sterilized in 70% (*v*/*v*) EtOH for 30 s, soaked in a 0.8% (*v*/*v*) NaOCl solution for 1 min, rinsed thoroughly with sterilized distilled water, and the outer layer of grains was peeled off. During the preparation of bacterial suspensions, the peeled grains were grown in darkness, in a liquid Murashige and Skoog (MS) containing 3% sucrose. One microgram of each construct was introduced into *Agrobacterium tumefaciens* (GV3101) competent cells using the freeze-thaw transformation method [46] and cultured in YEP agar Petri plates containing spectinomycin (50 mg L^−1^) and rifampicin (50 mg L^−1^). Single colonies cultured at 28 °C were inoculated into a 10 mL YEP suspension. An optical density (OD_600_) of 1.0 was obtained by overnight growth on a shaker at 150 rpm. The bacterial culture was then centrifuged at 3500 rpm for 15 min, the supernatant was discarded, and the pellet was resuspended in 10 mL of liquid MS with 3% sucrose supplemented with acetosyringone (Sigma Chemical Co. USA) at a concentration of 200 μM. The bacteria were cultivated for 1 h at 28 °C with shaking at 100 rpm. The OD_600_ of the bacterial suspension was adjusted to 0.6 using infiltration medium (MS with 3% sucrose, 50 mg/L acetosyringone, and 0.05% Silwet L-77), and the suspension was subsequently used for transformation. Immature seeds (35–40 days after heading) were immersed in the infiltration medium, and vacuum infiltration was repeated for 10 min. The infiltrated seeds were then placed on 1% MS solid medium plates, covered with foil, and incubated at 25 °C. Total protein was extracted from each grain [47] four days after infiltration [48,49], and the protein concentration was analyzed using the Bradford reagent (Bio-Rad, USA) at 595 nm. Protein (10 μg) was separated on a 4–15% gradient SDS-PAGE gel (Bio-Rad, USA), and expression was analyzed using anti-GFP (Abcam, UK) and anti-Phosphinotricin Acetyl Transferase (PAT) (Abcam, UK) antibodies using a chemiluminescence system, the LAS 4000 (GE Healthcare, IL, USA). The ratio of GFP to PAT was calculated using ImageJ, and three biological replicates were performed.

### 4.5. Glutenin Protein Analysis Using SDS-PAGE

Glutenin was extracted from single wheat kernels using a previously reported HMW-GS extraction protocol [9]. Protein (10 μg) extracted from grains of different varieties was separated using 10% SDS-PAGE and visualized using Coomassie Brilliant Blue R+250 staining solution (Bio-Rad, USA).

### 4.6. Breeding of NILs

To screen for genotypes carrying 1Bx7-overexpressing subunits, 607 wheat accessions from the National Institute of Crop Science (Wanju, Republic of Korea) were screened using specific PCR primers (Appendix A). The parental plants for breeding NILs were grown in a growth room with controlled temperature and photoperiod using modified speed-breeding methods [50]. Seeds were germinated at 4 °C for five days in a cold chamber (Dasol Scientific, Hwaseong-si, Republic of Korea) to break dormancy, and then vernalized at 8 °C under a 22 h photoperiod (LED bars, photosynthetic photon flux density, 15 cm above the floor, 150 μmol/m^2^/s) for four weeks in a growth room. After vernalization, growth room conditions were maintained at 22 °C under a 22 h photoperiod until harvest. Chisholm was used as the pollen parent to generate F1 plants. The local cultivars, Keumkang (1Ax2*, 1Bx7 + 1By8, 1Dx5 + 1Dx10) [44] and Chisholm, were grown side by side in a greenhouse for crossing. The recurrent parent, Keumkang, was backcrossed three times prior to two generations of self-pollination. The 43 bp insertion was confirmed using PCR markers in all backcross materials and self-pollinated lines. The gene duplication was confirmed in the BC_3_F_2_ and BC_3_F_3_ lines using TaBAC1215C06 markers (Appendix A). The BC_3_F_3_ progenies were analyzed using 10% SDS-PAGE and UPLC to detect different HMW-GS.

### 4.7. Genotyping of Segregating Populations and NILs

Genomic DNA was extracted from the tissue of seven-day old seedlings grown in Petri dishes at room temperature. The quality of the genomic DNA was assessed using a NanoDrop 1000 spectrophotometer (Thermo Scientific, MA, USA), and the integrity of the DNA was checked using 0.8% agarose gel electrophoresis. Polymerase chain reaction (PCR) amplification was performed in a thermal cycler (Applied Biosystems, MA, USA) with 20 μL reaction volumes for each sample containing 50 ng template DNA, 1 pmole of each primer, final 1× reaction buffer, 250 μΜ dNTPs, and 1 unit of Tag polymerase (Genetbio Prime DNA Polymerase, Korea) with initial denaturation at 94 °C for 5 min, followed by 35 cycles of 94 °C for 30 s, annealing at 55 °C for 30 s, and 72 °C for 30 s, with final extension at 72 °C for 5 min. The PCR products were electrophoresed in 1.2% agarose gels and visualized using a Davinch-K gel imaging system (Davinch-K, Seoul, Republic of Korea).

### 4.8. Glutenin Analysis Using UPLC

Glutenin analysis was performed by crushing one wheat seed, and a glutenin extraction method was performed with slight modifications [51]. The extracted glutenin was analyzed using an UPLC system (Alliance e2695, Waters Corp., MA, USA) with an ACQUITY UPLC Peptide BEH C18 column (300A, 1.7 μm, 2.1 mm × 50 mm) and a photodiode array detector. The mobile phases were H_2_O containing 0.1% trifluoroacetic acid (A) and acetonitrile containing 0.1% trifluoroacetic acid (B). The injection volume of the dissolved samples was 3 μL, and the flow rate was 0.55 μL/min. The solvent gradient was changed from 21% (B) to 47% (B) from 0 to 30 min, and the column and sample temperatures were set to 55 °C and 10 °C, respectively.

### 4.9. Protein Contents and SDS-Sedimentation Analysis by NIRS

The protein content and SDS-sedimentation value of 10 g of whole grains were determined using a NIRS (Foss, Denmark) calibration. The measured spectrum values were converted to a previously developed protein content (R^2^ = 93.6, root-mean-square error of cross-validation (RMSECV) = 0.67) and SDS-sedimentation (R^2^ = 94.3, RMSECV = 3.27) [37]. Measurements were repeated three times.

### 4.10. Seed Size Measurement

Seed size (length, width, and thickness) was measured with an ABSOLUTE Digimatic Caliper CD-20APX (Mitutoyo Corp., Kanagawa, Japan). Measurements were repeated three times.

### 4.11. Statistical Analysis

Statistical analysis was performed using Student’s *t*-tests to compare the three promoter experiments. Bars in all figures represent the mean ± SD determined from three biological replicates. Data were analyzed using one-way analysis of variance followed by Duncan’s multiple comparison test using R software (v3.5.1).

## 5. Conclusions

Three accessions (Tamaizumi, MT8195, and K2244) with only the 43 bp InDel in the promoter region and four accessions with a 1Bx7 gene duplication (MT8191, Chisholm, KS85WGRC01, and GR863) were selected and the protein levels were analyzed. Compared with the cultivars with only 43 bp inserted into the promoter, the expression of 1Bx7 was significantly increased in the cultivars with both the 43 bp insertion and *1Bx7* gene duplication. Chisholm (1Ax2*, 1Bx7^OE^ + 1By8*, and 1Dx5 + 1Dx10) was crossed with Keumkang (1Bx7 (1Ax2*, 1Bx7 + 1By8, and 1Dx5 + 1Dx10)), and 1Bx7^OE^ homozygous lines were selected from BC_3_F_3_ plants. These NILs showed a combination of 1Ax2*, 1Bx7^OE^ + 1By8*, and 1Dx5 + 1Dx10. The levels of 1Bx7 protein in the NILs were higher than those in Keumkang. In addition, the protein contents and SDS-sedimentation value were increased in NILs compared to Chisholm and Keumkang. This study is expected to contribute to the improvement of the quality of domestic wheat in Korea.

## Figures and Tables

**Figure 1 plants-12-01244-f001:**
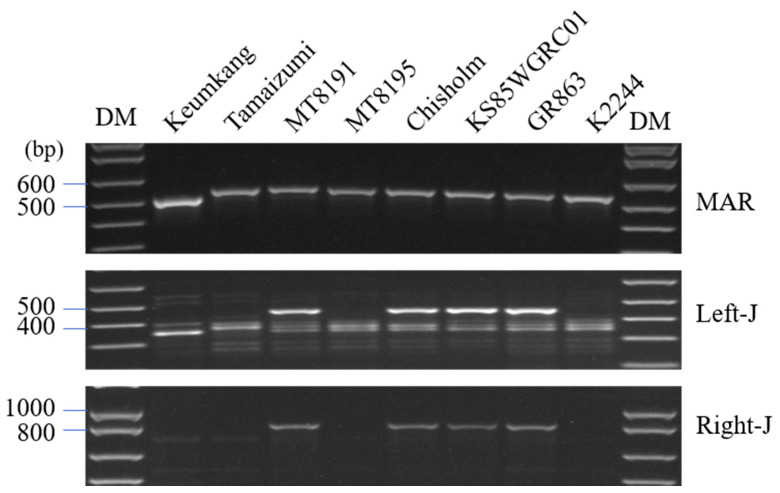
Selection of 1Bx7 and 1Bx7^OE^ wheat varieties. PCR products amplified with a 43 bp insertion marker (MAR) and left and right junction markers in different wheat varieties. DM, DNA size marker. The original gels are presented in Appendix A.

**Figure 2 plants-12-01244-f002:**
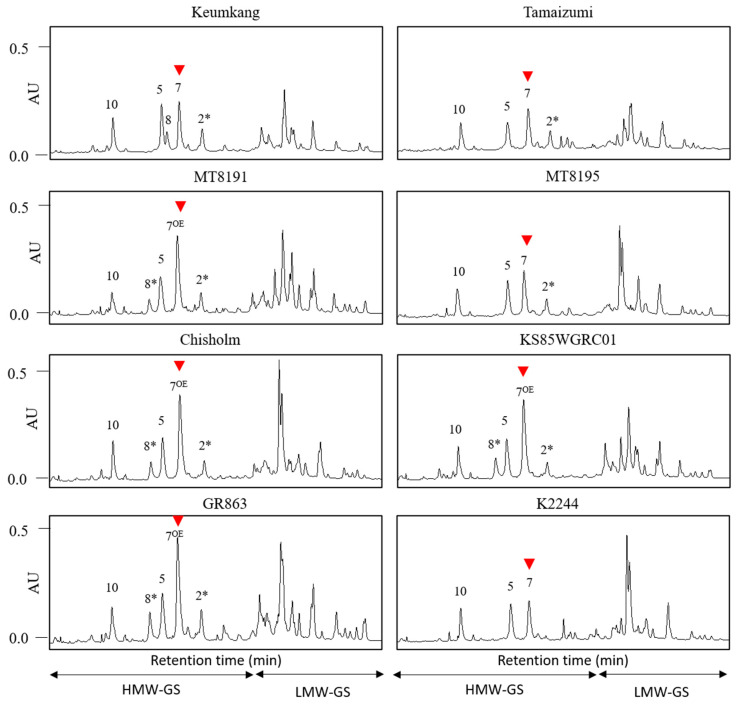
Measures of 1Bx7 and 1Bx7^OE^ expression using ultra performance liquid chromatography. Profiling of the proportion of HMW-GSs using extracts from the flour of Keumkang and seven other varieties using ultra performance liquid chromatography. AU, arbitrary units. HMW-GS, high-molecular-weight glutenin; LMW-GS, low-molecular-weight glutenin. The red arrows indicate *Glu-1Bx* alleles.

**Figure 3 plants-12-01244-f003:**
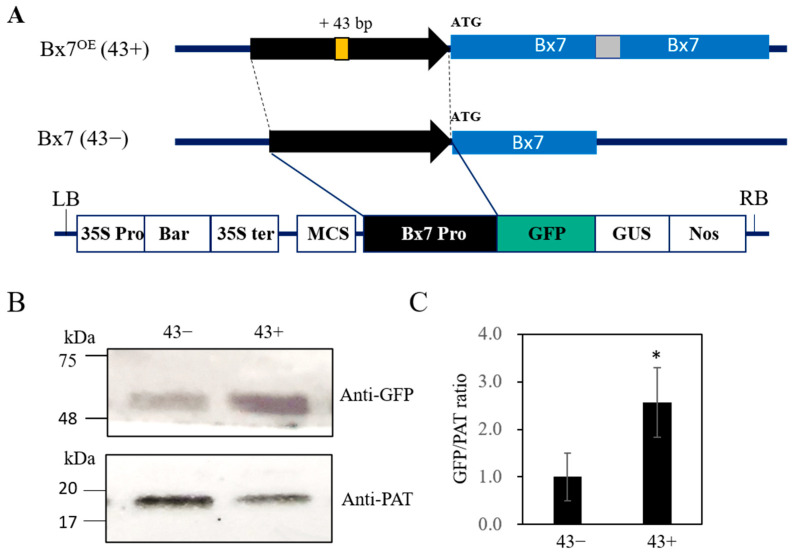
Construction of the 1Bx7 promoter harboring vectors and recombinant protein detection. (**A**) Schematic representation of the recombinant pBGFWFS7-pro1Bx7 vector; (**B**) GFP expression using a western blot; (**C**) Quantification of GFP expression. An anti-PAT antibody was used as an internal control. Error bars denote SD (*N* = 3). Asterisks indicate significant differences at * *p* < 0.05 as calculated using the Student’s *t*-test. The 43−, 43− promoter; the 43+, 43+ promoter; Bx7, *Glu-1Bx7*. The original blots are presented in Appendix A.

**Figure 4 plants-12-01244-f004:**
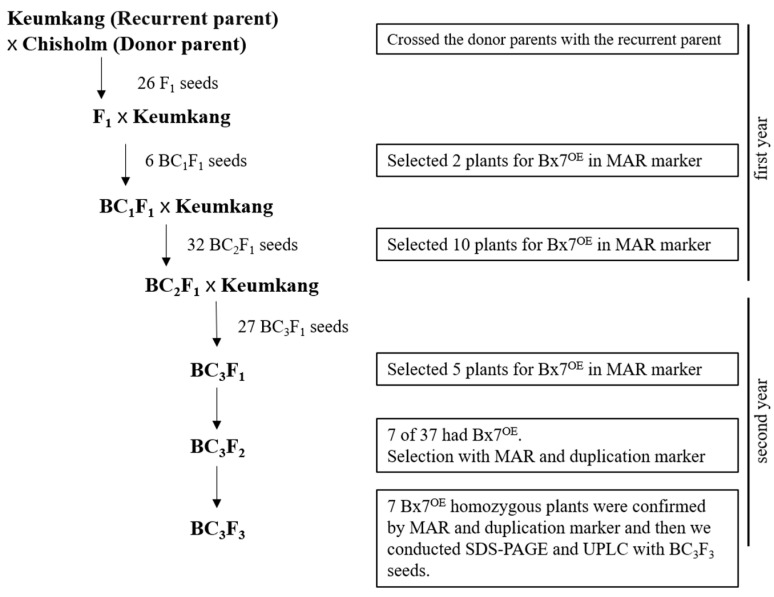
Crossing scheme for the production of the Keumkang near-isogenic lines.

**Figure 5 plants-12-01244-f005:**
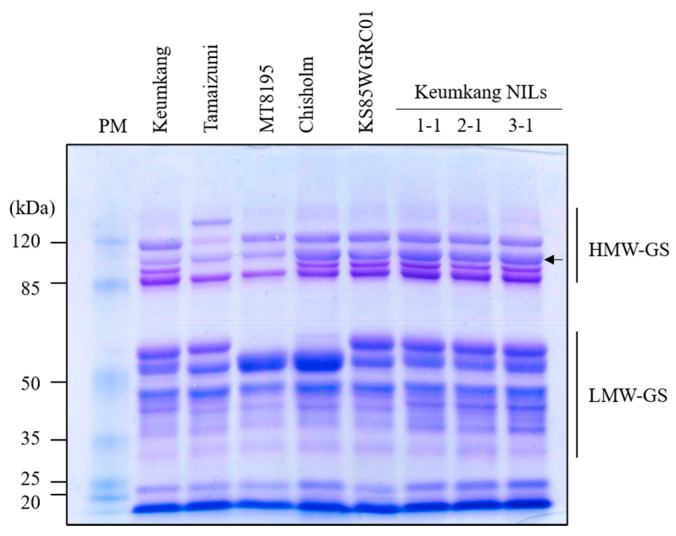
Protein levels of the 1Bx7 subunit in Keumkang (43−), 1Bx7 (43+), 1Bx7^OE^ cultivars, and NILs in SDS-polyacrylamide gel electrophoresis analysis. Arrow indicates Glu-1Bx7. PM, protein marker; HMW-GS, high-molecular-weight glutenin; LMW-GS, low-molecular-weight glutenin. The original gel is presented in Appendix A.

**Figure 6 plants-12-01244-f006:**
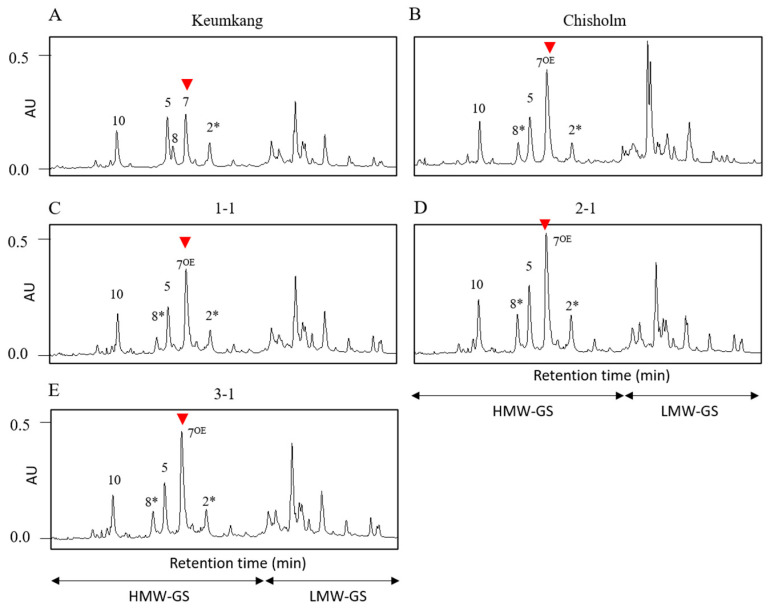
Identification of HMW-GS using ultra-performance liquid chromatography of Keumkang, Chisholm, and Keumkang near-isogenic lines. (**A**) Keumkang, (**B**) Chisholm, (**C**–**E**) NILs. AU, arbitrary units; HMW-GS, high-molecular-weight glutenin; LMW-GS, low-molecular-weight glutenin. The red arrows indicate *Glu-1Bx* alleles.

**Figure 7 plants-12-01244-f007:**
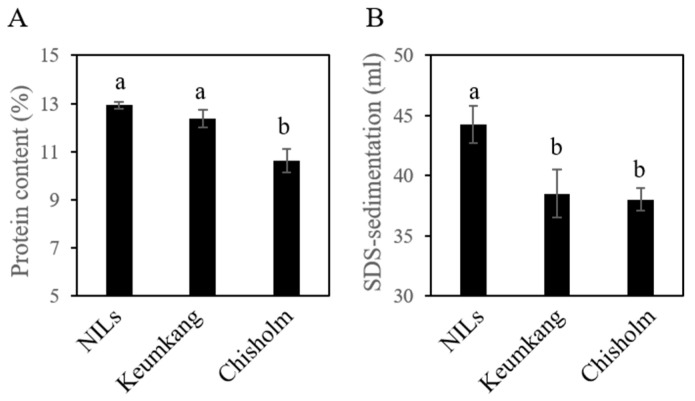
Protein contents and SDS-sedimentation analysis of NILs. (**A**) Protein contents and (**B**) SDS-sedimentation of Keumkang NILs, Keumkang, and Chisholm. Different letters on the bar indicate significant differences between treatments by Duncan’s multiple comparison test, *p* < 0.05.

**Table 1 plants-12-01244-t001:** Varieties containing a 43 bp insertion in the promoter and *1Bx7* gene duplication.

Variety	USDA-ARS ^1^ No.	NIAS IT No. ^2^	Country of Origin	43 bp Indel	Left/Right Junction	1Bx ^3^
KeumKang		IT213100	Republic of Korea	−	−	1Bx7
Tamaizumi	−	−	Japan	+	−	1Bx7
MT8191	Cltr17942	IT230937	USA	+	+	1Bx7^OE^
MT8195	Cltr17946	IT336848	USA	+	−	1Bx7
Chisholm	PI486219	IT230956	USA	+	+	1Bx7^OE^
KS85WGRC01	PI499691	IT230965	USA	+	+	1Bx7^OE^
GR863	PI508287	IT230969	USA	+	+	1Bx7^OE^
K2244	PI510693	IT341790	USA	+	−	1Bx7

^1^ U.S. Department of Agriculture-Agricultural Research Service; ^2^ registration number for plants in NARO Institute of Agrobiological Sciences national; ^3^ 1Bx, *Glu-1Bx*.

## Data Availability

Not applicable.

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
