# Peer review of "Generation of Wheat Near-Isogenic Lines Overexpressing 1Bx7 Glutenin with Increased Protein Contents and SDS-Sedimentation Values"

_plants, 2023, doi:10.3390/plants12061244_

Round 1

Reviewer 1 Report

The purpose of this research was to investigate the contribution that the high molecular weight 1Bx7OE might make to improve the bread making quality of domestic wheat grown in Korea.

Most, but not all, wheat varieties carrying the subunit have a 43 base pair (bp) indel in the promoter region. One problem was to decide whether the indel or duplicated allele was responsible for quality improvement.

To address this question 8 varieties with 1Bx7 (4) or 1BxOE were tested for presence of the indel; 7 carried it and only KeumKang lacked it. Since all three Glu-1 loci contribute to overall quality, three near-isogenic lines produced by backcrossing to KeumKang with Chisholm as donor were selected for common genotypes for subunits 1Ax1, 1BX7OE+1bx8 and 1Dx5+1Dy10 and differing from KeumKang by 1BX7OE and 1Bdx7. Flour samples were analysed by SDS sedimentation and protein content.

The protein content of the NILs was non-significantly higher that that of KeumKang and significantly higher than the 1BX7OE donor Chisholm. The SDS sedimentation values of the NILs were significantly higher than both parents.

The study showed:

1. both the 43 bp indel and 1Bx7OE subunit were necessary for improved quality

2. introduction of the 7Bx7OE subunit into Korean wheat varieties (a rare subunit in Korean varieties) would improve the bread making quality of domestic cultivars provided the 43 bp indel was also present.

Whereas the results are of particular interest to Korean wheat breeders and industry the first result provides support for an international issue that has been controversial in the past.

The paper needs considerable English editing, ‘CHISHOLM’ should be changed to ‘Chisholm’  throughout the paper

Some references should the edited to lowercase first letters.

A copy of the manuscript with hand-written suggested emendations is attached.

Author Response

The paper needs considerable English editing, ‘CHISHOLM’ should be changed to ‘Chisholm’  throughout the paper

Some references should the edited to lowercase first letters.

A copy of the manuscript with hand-written suggested emendations is attached.

Response:

Thank for your kind advise.

Corrections were made according to your advice.

Reviewer 2 Report

The study constructed NILs overexpressing 1BX7, which may be used in improvement of bread wheat. The work is interesting, but there are some problem need to be solved.

1)      How about the yield and agronomy traits, if the yield is very low, the usage may be limited. Pleased give the information of tillers, ear number, kernel number, grain weight, and other related parameters of NIL and parent lines.

2)      The protein content of NILs (12.94%) was 21.65% higher than that of CHISHOLM (10.63%) and was 4.54% higher to that of Keumkang (12.37%) but the difference was not significant. 21.65% higher, why the difference is not significant?

3)      Since overexpression of Glu-1Bx7 via allele 1Bx7OE contributes to high dough strength in some varieties, so the author should measure gluten content, farinograms to see if the dough was improved.

4)      Why the author choose to overexpress 1Bx7, not other subunits?

5)      Fig 2A is same with Fig 6A, why use the same Fig two times? Why not put them in one Fig?

6)      In Fig2 there are 8 lines, in Fig 5 some lines are not listed, why?

Author Response

The study constructed NILs overexpressing 1BX7, which may be used in improvement of bread wheat. The work is interesting, but there are some problem need to be solved.

1) How about the yield and agronomy traits, if the yield is very low, the usage may be limited. Pleased give the information of tillers, ear number, kernel number, grain weight, and other related parameters of NIL and parent lines.

Response:

Thank for your good comments.

We generated NILs by speed-breeding method so unable to measure agricultural traits due to the low number of seeds and growth condition. We are currently propagating NILs seeds, and plant to plant them in the field next year for mass production for agricultural trait and quality analysis. It is estimated that it will take two to three years for the results of the analysis to be released. In the next study, it is expected to show the results of agricultural trait and quality analysis.

2)  The protein content of NILs (12.94%) was 21.65% higher than that of CHISHOLM (10.63%) and was 4.54% higher to that of Keumkang (12.37%) but the difference was not significant. 21.65% higher, why the difference is not significant?

Response:

Thank you for your comment. It seems that the sentence was written incorrectly. What we want to mean is that the protein content of NIL is higher than CHISHOLM, but there is not much difference from Keumkang. Corrected the sentence.

Revised as below.

“The protein content of NILs (12.94%) was 21.65% higher than that of Chisholm (10.63%) and 4.54% higher to that of Keumkang (12.37%).” 

 3) Since overexpression of Glu-1Bx7 via allele 1Bx7OE contributes to high dough strength in some varieties, so the author should measure gluten content, farinograms to see if the dough was improved.

Response:

Thanks for your advice. We fully agree with you.

We are planning to propagate seeds and analyze quality, and plan to analyze quality in parent plants and NILs. However, since time is required for seed propagation and harvesting in the field, additional research is planned in the future, but it is difficult to show the analysis results at present. In the next study, it is expected to show the results of agricultural trait and quality analysis.

4) Why the author choose to overexpress 1Bx7, not other subunits?

Response:

In the previous references, cultivars with a high Glu-1 score were corelated in SDS-sedimentation value but also in other quality characteristics (sedimentation volume, gluten strength, dough tenacity, dough extensibility, elasticity index, grain hardness index, and farinograph stability). Glu-B1al (7OE+8) was evaluated to have higher gluten strength and farinograph dough stability than any other Glu-B1 allele. However, wheat cultivars containing 1Bx7OE are rare in Korea. Therefore, we introduced Bx7OE in domestic variety Keumkang to targeting high-quality strong flour.

5) Fig 2A is same with Fig 6A, why use the same Fig two times? Why not put them in one Fig?

Response:

Thanks for the advice.

Figure 2 compares the expression levels of Bx7 in selected cultivars, and Figure 6 compares the expression levels of Bx7 in NILs. Keumkang and Chisholm were used as controls, and in Figures 2 and Figure 6 look identical, but are not copied figure.

6) In Fig2 there are 8 lines, in Fig 5 some lines are not listed, why?

Response:

Figure 5 is a figure showing that Bx7 increases in NILs. Based on the results of Figures 1-2 and Table 1, two cultivars (Tamaizumi and MT8195) with only 43 bp inserted in the promoter and two cultivars (Chisholm, KS85WGRC01) with 43 bp inserted and duplicated Bx7 gene was selected as control groups, and HMW-GS subunits were compared with NILs. The number of cultivars was limited so that all could be displayed on an SDS-PAGE gel.

Reviewer 3 Report

Authors present a research article aiming to overexpress the allele 1Bx7 glutenin to improve HMW-GS, and consequently improve bread quality of Korean domestic wheat. The experimental design is well presented and logical; the manuscript is well written, however due to the fact that i'm not a native English-speaker, i can't evaluate the correctness. 

The manuscript can be accepted for publication in Plants, with minor comments to be taken into consideration:

1- All through the manuscript, authors used "expression levels" when describing protein accumulation. This is absolutely incorrect as a gene is expressed, but a protein is produced/accumulated. I would be glad to see that authors will change it for word such as accumulation, quantities or concentration

2- Figure 2: in the legend, authors described the abbreviation DM, but it is not present on the gel.

3- Table 1: the 1st column is indicated as "Accession name" however authors provided the Latine name of bread wheat (T. aestivum). So the description of the column is wrong and moreover that column is useless. The description of the 2nd column should be changed for "variety" instead of "name". Finally, the abbreviation USDA-ARS, NIAS-IT and 1Bx require a little more of description in the legend.

4- L118-120: reference [34] should be placed rather at the end of the sentence.

5- Figure 2: Authors did not provide in the legend the description of the red arrow shown on the peak 7. Why some peaks have a little star (asterisk)?

6- Figure 3: B) "GFP expression", please change accordingly to previous comment. On western-blot, you don't see gene expression but protein accumulation. Also, the right graph with ratio is not described in the legend of the figure. A) the figure is imperfectly composed. Please, improve.

7- L149: again replace "expression" by a different word, more appropriate to describe proteins

8 - Figure 6: again no description of the red arrow.

9- General comment to all figures: When statistics are performed and presented in the figure, the type of statistical test used should be described in the legend, even if described in the M&M. => figures have to be self explaining

10- L210: you can remove "however"

Author Response

Authors present a research article aiming to overexpress the allele 1Bx7 glutenin to improve HMW-GS, and consequently improve bread quality of Korean domestic wheat. The experimental design is well presented and logical; the manuscript is well written, however due to the fact that i'm not a native English-speaker, i can't evaluate the correctness. 

The manuscript can be accepted for publication in Plants, with minor comments to be taken into consideration:

1- All through the manuscript, authors used "expression levels" when describing protein accumulation. This is absolutely incorrect as a gene is expressed, but a protein is produced/accumulated. I would be glad to see that authors will change it for word such as accumulation, quantities or concentration.

Response:

Thank you for your advice. Manuscript was revised.

2- Figure 2: in the legend, authors described the abbreviation DM, but it is not present on the gel.

Response:

Thank for your advice.

Figure 2 was revised.

3- Table 1: the 1st column is indicated as "Accession name" however authors provided the Latine name of bread wheat (T. aestivum). So the description of the column is wrong and moreover that column is useless. The description of the 2nd column should be changed for "variety" instead of "name". Finally, the abbreviation USDA-ARS, NIAS-IT and 1Bx require a little more of description in the legend.

Response:

Thank for your advice.

Accession name column was deleted and ‘Name’ was changed to ‘Variety’.

USDA-ARS, NIAS-IT, and 1Bx were explained in legend.

4- L118-120: reference [34] should be placed rather at the end of the sentence.

Response:

Thank for your advice. It’s corrected.

5- Figure 2: Authors did not provide in the legend the description of the red arrow shown on the peak 7. Why some peaks have a little star (asterisk)?

 Response:

- Provided explanation for the red arrow in the figure legend.

- 2* and 8* are the name of Glu-1A and Glu-1D allele, respectively.

6- Figure 3: B) "GFP expression", please change accordingly to previous comment. On western-blot, you don't see gene expression but protein accumulation. Also, the right graph with ratio is not described in the legend of the figure. A) the figure is imperfectly composed. Please, improve.

Response:

Thank you for your comments.

Figure 3 legend has been revised.

7- L149: again replace "expression" by a different word, more appropriate to describe proteins

Response:

Thank you for your comments.

revised.

8 - Figure 6: again no description of the red arrow.

Response:

Provided explanation for the red arrow displayed at the top of the figure legend.

9- General comment to all figures: When statistics are performed and presented in the figure, the type of statistical test used should be described in the legend, even if described in the M&M. => figures have to be self explaining.

Response:

Statistical testing described in figure 3 and figure 7 legend.

10- L210: you can remove "however"

Response:

Revised.

Round 2

Reviewer 2 Report

The same fig can't be used two time, i suggest put Fig 2 and 7 together.

some agricultrue trait should be present, just compared with control.

Author Response

1. The same fig can't be used two time, i suggest put Fig 2 and 7 together.

Response:

Thank you for your suggestion.

I understood it to mean the UPLC results in Figures 2 and Figure 6.

We didn’t used same figure. We have included supporting data in additional file 1 to prove it. I considered merging the results of Figures 2 and 6, but the flow of the story is unnatural.

Please consider.

2. some agricultrue trait should be present, just compared with control.

Response:

Thank you for your advice.

There are currently few seeds, so we measured seed size of Keumkang, Chisholm, and NILs using a caliper and included in Supplementary data (Figure S4). As for the length and width of seeds, Chisholm was longer than Keumkang. The seed size and seed coat color of NILs were similar to those of Keumkang, the recurrent parent, than those of Chisholm.
